# Characterization of p53 Family Homologs in Evolutionary Remote Branches of Holozoa

**DOI:** 10.3390/ijms21010006

**Published:** 2019-12-18

**Authors:** Martin Bartas, Václav Brázda, Jiří Červeň, Petr Pečinka

**Affiliations:** 1Department of Biology and Ecology, University of Ostrava, 710 00 Ostrava, Czech Republic; bartasmartin@seznam.cz (M.B.);; 2Institute of Biophysics of the Czech Academy of Sciences, 612 65 Brno, Czech Republic; vaclav@ibp.cz

**Keywords:** p53, p63, p73, evolution, Holozoa

## Abstract

The p53 family of transcription factors plays key roles in development, genome stability, senescence and tumor development, and p53 is the most important tumor suppressor protein in humans. Although intensively investigated for many years, its initial evolutionary history is not yet fully elucidated. Using bioinformatic and structure prediction methods on current databases containing newly-sequenced genomes and transcriptomes, we present a detailed characterization of p53 family homologs in remote members of the Holozoa group, in the unicellular clades Filasterea, Ichthyosporea and Corallochytrea. Moreover, we show that these newly characterized homologous sequences contain domains that can form structures with high similarity to the human p53 family DNA-binding domain, and some also show similarities to the oligomerization and SAM domains. The presence of these remote homologs demonstrates an ancient origin of the p53 protein family.

## 1. Introduction

p53 is an intensively studied protein due to its role in tumor suppression, where p53 mutations are found in approximately 50% of human tumors [1,2,3] and dysregulation of p53 activity occurs in most other cancers [4]. It is accepted that p53 family proteins (p53, p63 and p73) arose from a common ancestral gene (most similar to contemporary p63 found in human [5]) and their functions diversified after gene duplications and rearrangements [6]. All three contemporary p53 family members are transcription factors that execute their functions through binding to DNA consensus sequences that are often located within locally structured regions [7,8,9,10]. The typical p53 consensus binding site has internal symmetry and two copies of the motif 5’-RRRCWWGYYY-3’ [11], although non-canonical half sites [12], targets with long spacers [9] and with various structural features [13,14] are also recognized by p63 and p73 [15]. All p53 family members act as tetramers through their oligomerization domains, and p63 and p73 also contain a C-terminal sterile alpha motif (SAM) [16,17], which has been lost in p53. The DNA binding domain (DBD) of p53 contains four of the five most highly conserved regions of vertebrate p53 and is a hotspot for p53 mutations in cancer, underscoring its importance for function [4,18].

The most detailed evolutionary studies focused on p53 family diversification after gene duplication and rearrangements, which led to the formation of proteins with different functional properties. These diversifications include variations in structurally disordered regions [19], in various secondary structures [20], loss of the SAM domain for p53 [21] and changes of regulatory phosphorylation sites [22]. On the contrary, the core domain containing the DBD retains high homology and preserves similar DNA binding affinities for all known p53 family proteins. The precise DNA-binding properties of these proteins are regulated not only by changes in this domain, but also in the N-terminal and C-terminal domains (especially for human p53) [23]. Post-translational modifications [24,25,26,27] as well as interaction with other proteins [28,29] modulates DNA-binding and subsequent transcriptional activation/suppression of individual target genes. 

While p53 family homologs are well described in Metazoa and especially in vertebrates [22], the two earliest branching homologs to date were described outside Metazoa in clade Choanoflagellates [30,31,32,33] and some p53 homologs were mentioned in some general studies of transcription factors in further premetazoan clades but did not go deeper into sequence analysis [34,35,36]. According to [6], the ancestral p53/63/73 protein present in early metazoans (like sea anemones) probably plays a role in germ cell protection. These homologs also have a tetramerization domain, which is present in today’s sea anemones (*Nematostella vectensis)* and is similar to the human tetramerization domain, except for missing a glutamine-rich region [20]. The biological role of the remote homologs found in Holozoa is unclear. To shed light onto the evolution of the p53 family, we have characterized the sequences of p53 orthologs in all non-animal eukaryotes, with a focus on the existing genomic and transcriptomic data of unicellular Holozoa. The investigation of sequence databases revealed the presence of p53 homologs in all clades of unicellular Holozoa (Choanoflagellatea, Filasterea, Ichthyosporea, Corallochytrea), with two new p53 homologs in *Ichtyophonus hoferi* and *Chromosphaera perkinsii* (both belong to Ichthyosporea). No p53 homolog was found outside Holozoa. We used amino acid sequence homology analyses and 3D modeling predictions to identify structural similarities in evolutionary close relatives and in human proteins. 

## 2. Results

### 2.1. Homology of the Sequences

Until now, most p53/63/73 homologs were found in the clade Metazoa. Using the Blast algorithm (blastp and tblastn) [37], we examined various databases (non-redundant protein sequences, WGS, EST, STS, GSS, TSA and non-annotated sets of protein sequences from http://multicellgenome.com/meet-our-organisms) [38,39]). The results revealed eleven significant hits outside Metazoa (E-value < 0.001), all designated as hypothetical proteins, five from clade Choanoflagellata (*Monosiga brevicollis* XP_001746020.1; *Monosiga brevicollis* XP_001747656.1; *Salpingoeca rosetta* XP_004994590.1; *Salpingoeca rosetta* XP_004991397.1 and *Salpingoeca rosetta* XP_004991396.1), one from clade Filasterea (*Capsaspora owczarzaki*, XP_004365382.2), four from clade Ichthyosporea (*Sphaeroforma arctica*, XP_014156832.1; *Creolimax fragrantissima* CFRG4869T1; *Ichthyophonus hoferi* Ihof_evm3s137; *Chromosphaera perkinsii* Nk52_evm78s1737) and one from clade Corallochytrea (*Corallochytrium limacisporum* Clim_evm153s157) (see Table 1, significant domain homology is indicated by a plus (+) mark). All eleven non-metazoan homologous sequences are given in Appendix A. To validate the homology of these proteins, we performed reciprocal searches using the phmmer tool [40] against the reference proteome database. These results show significant homology for multiple p53 family proteins, including human p53, p63 and p73 (Appendix A).

Although these proteins vary in size (170 to 768 aa residues) and in predicted pI (5.5 to 9.57), they all share significant homology with the p53 family DBD. Moreover, five have at least a partial tetramerization motif and four contain a SAM superfamily domain. None have homology with the p53 family transactivation domain (Figure 1). 

Due to the evolutionary distance and parallel evolution of these organisms for hundreds of millions of years, the large number of conserved sequences is surprising. We compared the presence of individual amino acids in the DBD of the six most remote p53 family homologs (from clades Filasterea, Ichthyosporea and Corallochytrea) with human p53 (Table 2). The 100% identity of 11 amino acid residues and conservation of positively charged residues at positions 248 and 273 of human p53 points to their crucial importance for p53 family protein structure and function. Fascinatingly, these conserved residues are important for p53-DNA binding in human p53 and/or are hotspot mutation sites in human cancer. 

We inspected the exon-intron structure of p53 homologs in two paralogs of *Monosiga brevicollis*, three paralogs of *Salpingoeca rosetta*, one homolog from *Capsaspora owczarzaki* and one from *Sphaeroforma arctica* (Figure 2). All these homologous genes are very short, maximum length just over 5 kbp in *Salpingoeca rosetta* (XM_004994533.1; protein XP_004994590.1). For comparison, exon-intron structure of three close relative Metazoans are shown and it is evident that the length of introns in p53 homologs increases with the organism’s complexity (for example, the human p63 gene is over 250 kbp long). Furthermore, it is interesting that the homologous gene in *Capsaspora owczarzaki* has only two exons, so the whole DBD is located in exon 2. We could not investigate exon-intron structure of homologous genes from more distant clades, because their genomic DNA sequences are unavailable at this time.

### 2.2. Homology of Predicted Structures

We used QUARK ab initio protein structure prediction algorithm [46,47] to de novo model the structure of the DBDs. It is clearly visible that all Holozoan p53 family homologs contain functionally important beta sheets in their DBDs (as well as human p63 ab initio modelled structure and most importantly also experimentally verified p53 family DBDs), but the alleged homolog from *Entamoeba histolytica* contains mainly alpha-helices (Figure 3A). Root mean square deviation (RMSD) between experimentally determined structure of human p63 DBD (PDB code: 2rmn) and our ab initio modelled human p63 DBD was only 1.259 Å. Using SWISS-MODEL [48] and UCSF Chimera [49] we visualized 3D models of their DBDs (Figure 3B). Overall RMSD was 0.718 Å (compared to the experimentally determined DBD structure of human p63 as the reference structure). Since the accepted cut off for similarity is <2 Å, this points to a high level of DBD structural conservation in these distant homologs. Simulation of the *E. histolytica* protein was not allowed due its low homology. All predicted models in PDB format are enclosed in Appendix A (QUARK) and Appendix A (SWISS-MODEL). The fact that the sequences in the DBD might arrange in a similar tridimensional conformation as human p53 suggests these orthologs in unicellular Holozoa could bind to DNA, as observed for animal p53/63/73.

The most distant predicted structure of p53 family homolog DBDs was found in *Corallochytrium limacisporum*. This is a single-celled eukaryote living in coral reef lagoons; it was considered to be part of fungi, but contemporary phylogenetic analyses based on DNA sequencing show that this organism is a member of the Holozoa branch [38], as are all known p53 family homologs. Structure-based sequence alignment of primary amino acid sequences of human p63 and the six most distant non-Metazoan organisms (Figure 4) shows high homology (orange rectangles), complete identity is highlighted in the consensus line in red upper case. Structure-based sequence alignment show us combination of sequence alignment and structural features (alpha helices, beta sheets) as determined by the Match Maker tool (3D superimposition of all template-based structures predicted and experimental 2rmn p63). 

## 3. Discussion

The high homology between the core domains of p53 family members suggests that the remote proteins found, containing newly described homologs, have DNA binding features similar to p53 from mammals, amphibians, fish, insects and nematodes [50] (Table 2, alignment of the entire DBD protein sequences of novel homologs with human p53 is shown in Appendix A). The positive charge of the key DNA binding amino acid residues 248 and 273 of the human canonical p53 sequence is conserved in all remote homologs; interestingly these residues are always arginines (as in the human sequence) or lysines, or a combination of the two. Furthermore, two of four zinc-coordinated amino acid residues, particularly C176 and C238, are 100% conserved in all remote homologs. Zinc ions play a critical role in stabilizing the architecture of the DBD. We also found significant homology in the tetramerization domains, which indicates the possibility of functional tetramers forming in these homologs. Interestingly, tetramerization domains were not found in *Monosiga brevicollis*, *Chromosphaera perkinsii* and *Corallochytrium limacisporum*, but this may be due to the fact that the tetramerization domain is relatively short and homology lies slightly below selected E-value threshold. Obviously, the binding properties of the remote homologs should be determined by combinations of wet-lab methods.

From our data, we may speculate as to the original function of p53 homologs in unicellular Holozoans and Metazoans. It is supposed that the first evolutionarily role of p53 in primitive Metazoans could be in apoptosis regulatory network via activation by upstream kinase CHK2 [30] and/or in DNA repair via activation of RNR (ribonucleotide reductase) gene expression, which produces deoxyribonucleotide triphosphates (dNTPs) required for DNA replication and repair [30,51]. The absence of an homologous transactivation domain in the most remote p53 family homologs is consistent with previous observations that the transactivation domain first appeared in Placozoa [52]. However, one must be aware that contemporary databases and articles contain misleading information. For example, it is clear that there is no p53 family homolog in plants, but there is a misleading GenBank annotated p53 protein sequence in *Zea mays* (GenBank: AAT42177.1). This is a clear artefact due to human DNA contamination (the *Zea mays* sequence has 100% identity with human p53). The presence of a p53 family homolog has also been reported in *Entamoeba histolytica* (Amoebozoa group) [53]–however our results indicate that this protein is not a p53 family homolog; the protein shows the highest homology to the Alpha kinase superfamily (E-value = 2.5 × 10^−8^) and lacks homology with the p53 family DBD domain (Appendix A, compared to thousands of significant hits for the proteins characterized in this study including human p53, p63 and p63 proteins in Appendix A).

The finding of an alleged p53 family homolog in *Entamoeba histolytica* provides a good example of the problems inherent in searching for distant protein homologs based solely on overall sequence similarity and has led to misleading information in subsequent papers, including a well-known review on p53 family evolution, where Amoebozoa is included in the evolutionary tree [30]. To avoid such problems, we initially used sequence similarity of the core domain, the most highly conserved region in p53 family evolution and known to be vital for functional activity. We then used structural predictions to model the 3-D organization of the putative homologs and identify those with similar structures. Based on these recent data we have updated the p53 family ancestral tree and show the closest evolutionary branches where p53 family homologs are not present (Figure 5).

Recently, two paralogs from *Monosiga brevicollis* and three paralogs from *Salpingoeca rosetta* were mentioned in a paper dealing with the p53 gene family in vertebrates [54]. These results support our data from non-Metazoan analyses. Our results confirm the presence of p53 family homologs in remote members of Holozoa clades Filasterea, Ichthyosporea and Corallochytrea. All newly characterized homologs contain a domain predicted to have high structural similarity to the core DNA-binding domain of p53 family proteins. Moreover, some of these unicellular organisms also contain a tetramerization motif and/or a SAM superfamily domain. We also analyzed exon-intron structures where the genomic sequences were available. It is of interest that the whole DBD of *Capsaspora owczarzaki* is located inside a single exon, suggesting that this may be the ancestral form of gene arrangement. Unfortunately, genomic sequencing data for the most remote branches (Ichthyosporea and Corallochytrea) are not available, so we cannot infer if the *Capsaspora owcarzaki* p53 homolog contains a similar exon-intron arrangement to the ancient form of the gene. As depicted in Table 1, only the p53 DBD is conserved in all remote homologs, and it is well known that this part of the protein has the critical biological function of binding to DNA. The N-terminal transactivation domain first appeared in Metazoa, as shown by Aberg et al. [52]. The tetramerization motif is too short and variable to be analyzed from the phylogenetic point of view and SAM domains are present in many proteins through all three domains of life (Eukaryota, Archaea, Bacteria) [55], so they are not specific to p53/63/73 evolution. Large bioinformatic analyses suggest that gain of p53 DBD occurred together with the gain of Runt, zf-C2HC and bZIP_Maf and the loss of WRKY transcription factors [35]. The results of this study characterize in detail the presence of p53 family homologs in unicellular evolutionary distant branches of Holozoa and provide important insights for further research on this key family of transcription factors.

## 4. Materials and Methods

### 4.1. P53 Family Homolog Search

The human p53 canonical protein sequence (UNIPROT ID: P04637-1) was used for blast search against the contemporary version of the Reference protein database (refseq): https://blast.ncbi.nlm.nih.gov/Blast.cgi?PAGE = Proteins [37], excluding taxid Metazoa (taxid:33208). To broaden our search we downloaded non-annotated sets of protein sequences (without GenBank IDs) from the database http://multicellgenome.com/meet-our-organisms [38,39]. Using UGENE [56] we organized a local blastp database and searched using the newly identified p53 family sequence from *Sphaeroforma arctica*. This approach identified four additional p53 family homologs. All eleven non-metazoan p53 homologous sequences are enclosed in Appendix A. We also used non-redundant protein sequences, EST, WGS, STS, GSS, TSA and INTERPRO databases to validate and confirm our results. No other homologs have been found in these databases.

### 4.2. Validation of p53 Homologs (Reciprocal Search)

All putative p53 homologs were reverse searched using the sensitive tool phmmer (https://www.ebi.ac.uk/Tools/hmmer/search/phmmer) [40] against “Reference Proteomes” database, E-value cut-off set to 1. Detailed results are enclosed in Appendix A.

### 4.3. Isoelectric Point Predictions

Isoelectric point predictions were made using IPC—Isoelectric Point Calculator (http://isoelectric.org/index.html) [57].

### 4.4. Domain Searches and Visualization

Domain annotation was performed using NCBI Web-CD (https://www.ncbi.nlm.nih.gov/Structure/bwrpsb/bwrpsb.cgi) with default parameters except for “search mode” (live search only) and “expect value threshold” (0.001). Results were visualized with the “browse results” option and “compact mode” view [58], *p*-values of all domain hits are enclosed in Appendix A.

### 4.5. Nuclear Localization Signal Predictions

Nuclear localization signal predictions were made using cNLS Mapper (http://nls-mapper.iab.keio.ac.jp/cgi-bin/NLS_Mapper_form.cgi) [59] with cut-off score = 3 (“localized to both the nucleus and the cytoplasm”).

### 4.6. Estimation of Evolutionary Age

Information on the evolutionary age of particular phylogenetic nodes was mined from TIMETREE database (http://www.timetree.org/) [60,61]. Detailed figures with 95% confidence intervals are enclosed in Appendix A.

### 4.7. Validation of Entamoeba Histolytica Putative p53 Homolog

Web version of phmmer algorithm was used [40], sequence database was set to “Reference Proteomes”, E-value cut-off set to 1. As a query protein sequence, DBD from putative Entamoeba histolytica p53 homolog was used (FASTA file enclosed in Appendix A). To demonstrate that this putative p53 homolog contains another domain (unrelated to p53 family), SMART tool [62] with PFAM domains search option was used.

### 4.8. Alignment of DBD from the Most Distant Human p53 Homologs

Alignment was done with MUSCLE [63] under default parameters (flow was used [56]), FASTA sequences of p53 DBD are enclosed in Appendix A. Conserved human p53 DBD regions II–V were added according to the graph published previously [64].

### 4.9. p53 Protein Tree

The p53 protein tree is based on eleven unicellular non-Metazoans and twenty-seven other representative closest relative Metazoan species (enclosed in Appendix A), which were previously aligned by MUSCLE [63], was built using MrBayes [65] with the following parameters: Likelihood model (Number of substitution types: 6 GTR; Substitution model: Dayhoff-protein; Rates variation across sites: gamma + invariable); Markov Chain Monte Carlo parameters (Number of generations: 10,000; Sample a tree every 10 generations); Discard first 250 trees sampled. Resulting p53 protein tree is enclosed in Appendix A.

### 4.10. Modelling of 3D Protein Structures

First, we have taken DBD sequences from the most distant holozoan p53 family homologs (clades Filasterea, Ichthyosporea and Corallochytrea) and modelled 3D structures ab initio using QUARK webserver (https://zhanglab.ccmb.med.umich.edu/QUARK2/) [46,47]. Then we used SWISS-MODEL template-based approach (https://www.swissmodel.expasy.org/interactive) [48] to better predict 3D structures using individual FASTA sequences and reference PDB:2rmn as the solution structure of p63 DBD from *Homo sapiens* [66]. All resulting PDB files are enclosed in Appendix A. Predicted structures of p53 family homologs were visualized in UCSF Chimera 1.12 [49,67]. Structural alignment of human and most distant unicellular Holozoan DBDs was made using MatchMaker [67].

### 4.11. Comparison of Exon-Intron Composition of the Selected p53 Homologs

Complementary DNAs (cDNAs) and related genomic regions were downloaded from NCBI and curated as two FASTA files (separate FASTA file for all cDNAs and separate FASTA file for all corresponding genomic regions). These FASTA files were uploaded to Gene Structure Display Server (http://gsds.cbi.pku.edu.cn/) [68] and DBDs were annotated using tabular file manually filled with PFAM data on all inspected cDNAs obtained previously. The DNA tree was reconstructed using Phylogeny.fr [69] webserver and “A la carte” mode of analysis with T-Coffee aligner [70] selected for alignment of cDNAs and MrBayes [65] phylogeny with subsequent parameters; Likelihood model: Number of substitution types “GTR”, Substitution model “Default”, Rates variation across sites “Invariable + gamma”, Markov Chain Monte Carlo parameters “10 000 generations and sampling tree every 10 generations”, Discard first “250” trees sampled. The detailed resulting DNA tree is enclosed in Appendix A.

## Figures and Tables

**Figure 1 ijms-21-00006-f001:**
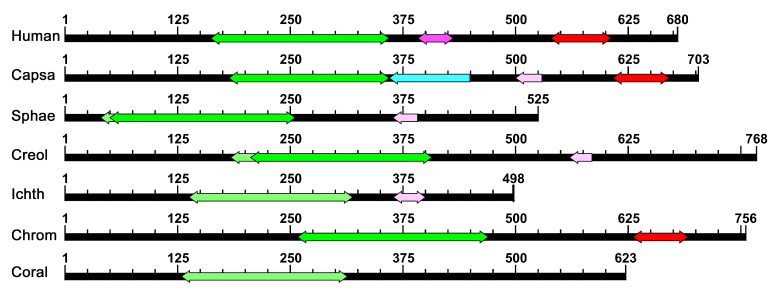
The most evolutionarily distant homologs of human p53 family proteins. Particular domains are highlighted (green and light green for p53 superfamily, purple and light purple for the tetramerization motif, red for SAM domain and light blue for sporulation related domain found only in *Capsaspora owczarzaki*). Lighter shades symbolize lower levels of homology. Abbreviations: Caps (*Capsaspora owczarzaki*), Sphae (*Sphaeroforma arctica*), Creol (*Creolimax fragrantissima*), Ichth (*Ichthyophonus hoferi*), Chrom (*Chromosphaera perkinsii*), Coral (*Corallochytrium limacisporum*).

**Figure 2 ijms-21-00006-f002:**
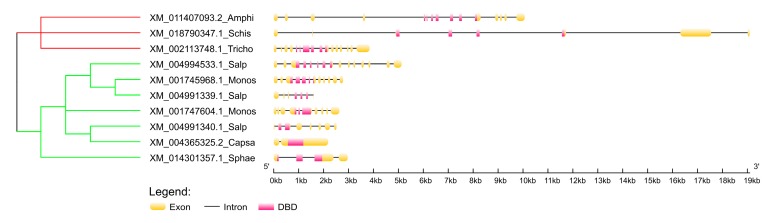
Exon-intron structure of remote human p53 family homologs in the earliest branches of Holozoa. Non-metazoan branches are colored green and metazoan branches red. The tree is based on DNA alignment of cDNAs (T-Coffee aligner) and MrBayes phylogeny. Detailed tree with bootstrapping values and real branch lengths is enclosed in Appendix A. Abbreviations: Amphi (*Amphimedon queenslandica*), Schis (*Schistosoma mansoni*), Tricho (*Trichoplax adhaerens*), Salp (*Salpingoeca rosetta*), Monos (*Monosiga brevicollis*), Capsa (*Capsaspora owczarzaki*), Sphae (*Sphaeroforma arctica*).

**Figure 3 ijms-21-00006-f003:**
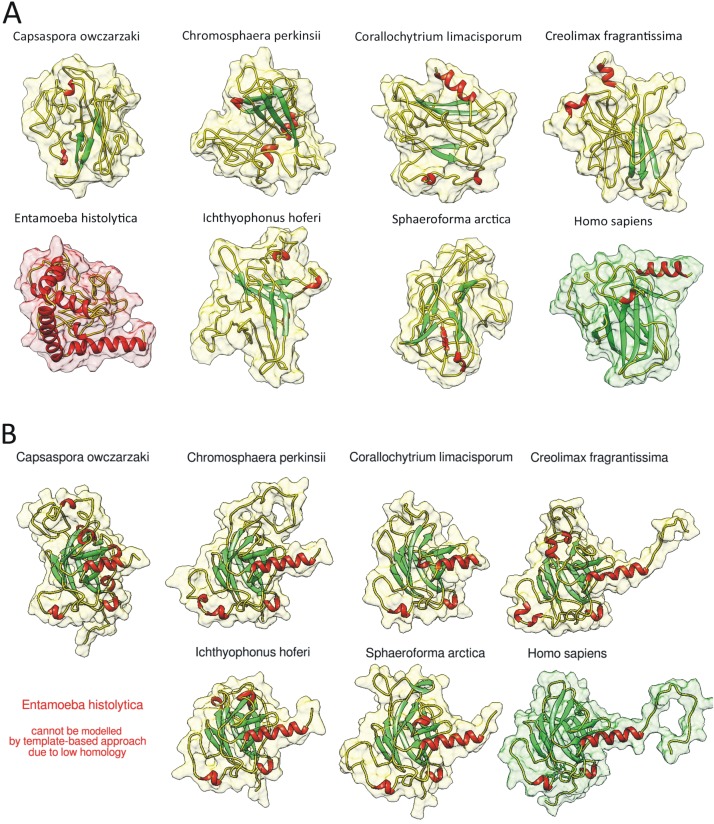
(**A**) Ab initio modelled DBD structures using the QUARK algorithm [45,46] of the most distant unicellular Holozoa, the alleged homolog from *Entamoeba histolytica* and human p63 DBD (also modelled ab initio from its sequence, to be comparable). (**B**) The most distant unicellular Holozoan homologs of human p53 family DBDs and experimentally validated p63 DBD. Template based predictions were made using NMR solution model of human p63 DBD (PDB: 2rmn). Beta sheets are colored green, alpha helixes red.

**Figure 4 ijms-21-00006-f004:**
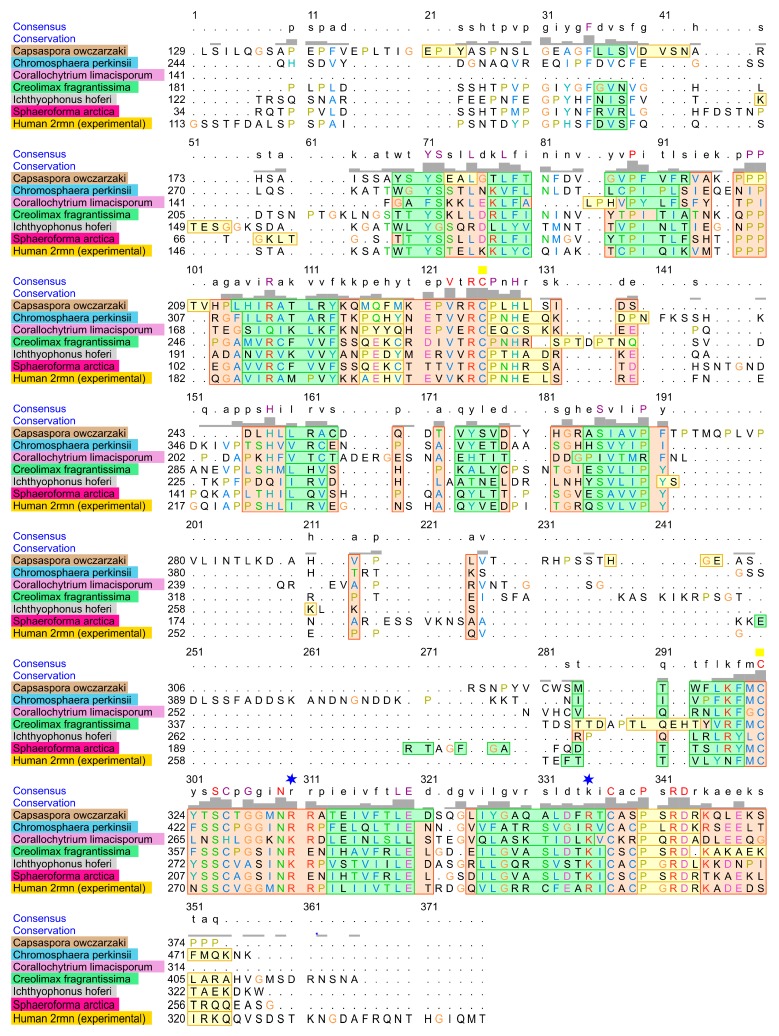
Amino acid alignment of the human p63 DBD and the homologous DBDs from the six most distant non-Metazoan organisms; identical residues are shown above the alignment; pink shaded areas represent helixes, green represents beta sheets and yellow represents unstructured parts, orange rectangles show sites of high structural similarity. Text colors represent amino acid similarity. Two blue stars depict positions of the most important DNA binding residues R248 and R273 (related to positions in human p53 sequence). Two yellow squares indicate positions of zinc-coordinating amino acid residues C176 and C238 (related to positions in human p53 sequence).

**Figure 5 ijms-21-00006-f005:**
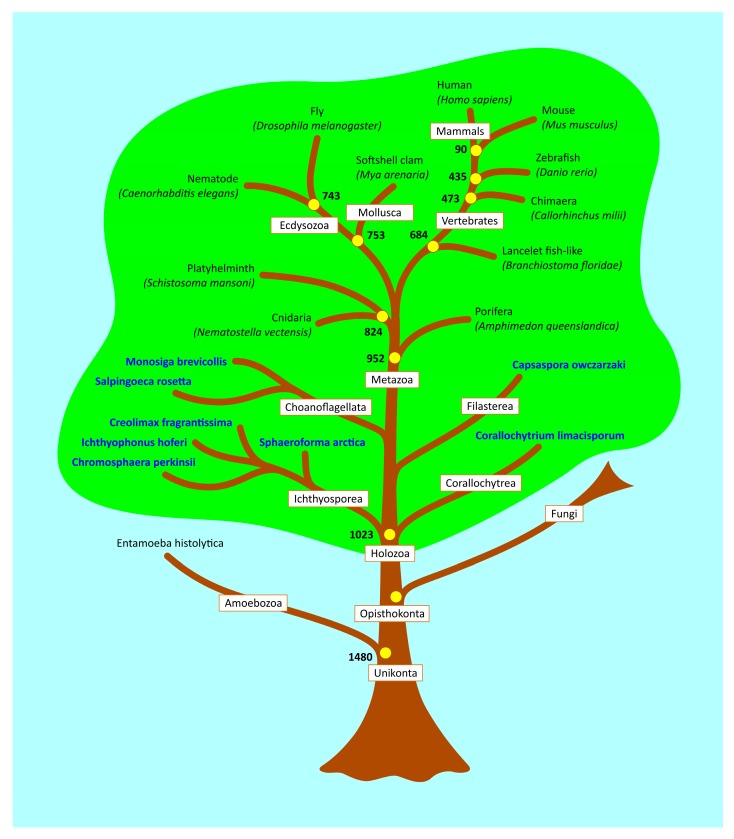
Simplified p53 family tree with our newly characterized p53 family homologs. Yellow circles show the evolutionary age of phylogenetic nodes by TIMETREE database. Non-metazoan organisms with p53 family homologs are in bold. The p53 family world is highlighted in green and the non-p53 family world (including Amoebozoa and Fungi) is in blue.

**Table 1 ijms-21-00006-t001:** List of p53 family homologs in unicellular organisms belonging to Holozoa. “−” absence and “+” presence of particular protein features (E-value < 0.001). Abbreviations: TAD (transactivation domain), DBD (DNA-binding domain), TET (tetramerization motif), SAM (sterile alpha motif), NLS (nuclear localization signal).

Organism Name	ID of p53 Homolog	PFAM Domains				NLS
	Protein Length (aa)	TAD	DBD	TET	SAM	
	Predicted pI					
*Monosiga brevicollis* (class Choanoflagellate)	XP_001746020.1	−	+	−	−	+
	523					
	6.68					
	XP_001747656.1	−	+	−	−	+
	571					
	5.89					
*Salpingoeca rosetta* (class Choanoflagellate)	XP_004991396.1	−	+	−	−	−
	170					
	5.55					
	XP_004991397.1	−	+	−	+	+
	351					
	5.74					
	XP_004994590.1	−	+	+	+	−
	613					
	6.72					
*Capsaspora owczarzaki* (class Filasterea)	XP_004365382.2	−		+	+	+
	703					
	7.05					
*Sphaeroforma arctica* (class Ichthyosporea)	XP_014156832.1	−	+	+	−	−
	525					
	9.57					
*Creolimax fragrantissima* (class Ichthyosporea)	CFRG4869T1	−	+	+	−	−
	768					
	8.72					
*Ichthyophonus hoferi* (class Ichthyosporea)	Ihof_evm3s137	−	+	+	−	+
	498					
	7.75					
*Chromosphaera perkinsii* (class Ichthyosporea)	Nk52_evm78s1737	−	+	−	+	+
	756					
	6.29					
*Corallochytrium limacisporum* (Corallochytrea)	Clim_evm153s157	−	+	−	−	+
	623					
	5.50					

**Table 2 ijms-21-00006-t002:** Most conserved amino acid residues in the DBD of the six most remote p53 family homologs related to human p53. For detailed alignment with consensus positions see Appendix A.

Human p53 Position	Amino Acid	Conservation in All Homologs	Function
142	P	100%	Unknown
173	V	100%	V173L hotspot [41]
175	R	100%	R175H hotspot [42]
176	C	100%	Zn^2+^ binding [43]
238	C	100%	Zn^2+^ binding [43]
241	S	100%	S241F hotspot [42]
247	N	100%	N247I hotspot [41]
248	R	+charged aa	R248W hotspot [42]
273	R	+charged aa	R273H hotspot [42]
275	C	100%	Unknown
278	P	100%	Unknown
280	R	100%	R280K hotspot [44]
281	D	100%	D281G hotspot [45]

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
