# Peer review of "Characterization of p53 Family Homologs in Evolutionary Remote Branches of Holozoa"

_ijms, 2019, doi:10.3390/ijms21010006_

Round 1
Reviewer 1 Report
MS TITLE: Characterization of p53 family homologs in evolutionary remote branches of Holozoa
AUTHORS: Martin Bartas, Václav Brázda, Jiří Červeň and Petr Pečinka
The authors have presented a detailed characterization of p53 family homologs in remote members of the Holozoa group. They have updated the p53 family ancestral tree and show the closest evolutionary branches where p53 family homologs are not present.
Overall, the manuscript is clear and well written. Here are some suggestions of corrections/improvements:
- page 3, line 21: … see Table 1 – significant domain homology is shown in green ?
- page 7, 2nd paragraph: angstroms or Å ?
- page 16, Data availability - … from the corresponding authors upon reasonable (?!) request.
- pages 18, 19, References: citations no 39, 43, 52 are incomplete.
Author Response
Thank you for your positive feedback and recommendations, we have fixed all of these issues in change tracking mode
Reviewer 2 Report
The manuscript by Bartas M. and co-authors address the evolution of the p53 family homologs of remote branches of Holozoa. The authors applied appropriate bioinformatic and structure prediction methods to characterize in detail p53 family homologs. The main conclusion is that remote homologs exhibit an ancient origin of the p53 protein family. The results are clearly presented but there is lack of a wider discussion about the biological meaning of the obtained results and their significance. It is also worth mentioning whether a similar evolutional pathway outside Metazoa has been already characterized for other transcriptional factor genes. Some kind of comparison, if data are available, would situate the results in broad context of evolution of the transcription factors.
Detail comments:
Table 1. “Y” could be replaced by ‘’+”. It seems to be more appropriate.
Figure 3. The quality of figure 3 might be better.
Author Response
Thank you for your comments. We have broadened the Introduction and Discussion to cover the suggested topics more fully.
We have followed the recommendation in Table 1 and Figure 3 has been submitted in high resolution. If necessary, the editorial office would be able to provide the original figure files.
Reviewer 3 Report
The manuscript “Characterization of p53 family homologs in evolutionary remote branches of Holozoa” offers a comprehensive characterization of p53 homologs in unicellular holozoa based on bioinformatics analyses of available genome sequencing data, as well as structural predictions of their DNA binding domains.
Given the consensus view that the function of the ancestral p63/p73-like gene in early metazoans was to protect the germline from DNA damage, the apparent presence of p53-related sequences in the genomes of unicellular organisms is an interesting observation and raises the question about the function of the encoded proteins.
While the comparative genomic analyses, as well as the in silico structural predictions done by the authors are thoroughly executed, the identification of p53-like sequences in unicellular organisms, as the authors themselves indicate, is not an entirely novel finding. Earlier reports have pointed out the presence of potential p53-like proteins (or at least sequences with high similarity to the p53 DBD) in Monosiga brevicollis (Nedelcu and Tan, 2007; King et al., 2008) and Capsaspora owczarzaki (Sebe-Pedros et al., 2011).
The current manuscript extends these earlier findings to additional organisms, indicating the presence of p53-like proteins in at least some pre-metazoan lineages. The structural predictions of the DBDs suggest similar DNA binding activity of these proteins compared to p53; however, this is somewhat speculative.
The paragraphs in the discussion regarding the Zea Mays genome annotation or the apparent misidentification of an E. histolytica p53-like protein should be shortened or deleted. Instead, the putative function(s) of the newly identified p53-like proteins should be discussed in more detail.
The introduction should provide more information on p53 evolution and function (activation by DNA damage, germ-line protection etc,).
Author Response
I would like to take this opportunity to thank the referees for their time and effort with our manuscript. We have been able to address all of their comments and questions in the revised manuscript, which has improved the paper and we hope that this updated version is now acceptable for publication. Detailed responses to the reviewers’ comments are provided in bold.
Reviewer #3
The manuscript “Characterization of p53 family homologs in evolutionary remote branches of Holozoa” offers a comprehensive characterization of p53 homologs in unicellular holozoa based on bioinformatics analyses of available genome sequencing data, as well as structural predictions of their DNA binding domains. Given the consensus view that the function of the ancestral p63/p73-like gene in early metazoans was to protect the germline from DNA damage, the apparent presence of p53-related sequences in the genomes of unicellular organisms is an interesting observation and raises the question about the function of the encoded proteins. While the comparative genomic analyses, as well as the in silico structural predictions done by the authors are thoroughly executed, the identification of p53-like sequences in unicellular organisms, as the authors themselves indicate, is not an entirely novel finding. Earlier reports have pointed out the presence of potential p53-like proteins (or at least sequences with high similarity to the p53 DBD) in Monosiga brevicollis (Nedelcu and Tan, 2007; King et al., 2008) and Capsaspora owczarzaki (Sebe-Pedros et al., 2011).
Yes, we agree; we stated in our manuscript that some of these p53 homologs have already been shown – and we cited the mentioned papers in our manuscript.
The current manuscript extends these earlier findings to additional organisms, indicating the presence of p53-like proteins in at least some pre-metazoan lineages. The structural predictions of the DBDs suggest similar DNA binding activity of these proteins compared to p53; however, this is somewhat speculative. The paragraphs in the discussion regarding the Zea Mays genome annotation or the apparent misidentification of an E. histolytica p53-like protein should be shortened or deleted. Instead, the putative function(s) of the newly identified p53-like proteins should be discussed in more detail.
Thank you for these positive comments. We have shortened this paragraph. However, it is important to point out that some publicly accessible data are not correct. As suggested, we have broadened the part about putative functions of the newly characterized p53 homologs.
The introduction should provide more information on p53 evolution and function (activation by DNA damage, germ-line protection etc,).
Thank you. We have broadened this part of the introduction to include more information on the suggested function of p53 homologs in evolutionary distant organisms.